# Unmixed Diet Versus Total Mixed Ration in Beef Cattle Fed High-Concentrate Diets: Effects on Methane Emissions, Animal Performance, and Rumen Fermentation

**DOI:** 10.3390/ani15050723

**Published:** 2025-03-03

**Authors:** Amira Arbaoui, Gonzalo Gonzalo, Alejandro Belanche, Antonio de Vega

**Affiliations:** 1Departamento de Producción Animal y Ciencia de los Alimentos, Instituto Agroalimentario de Aragón (IA2), Universidad de Zaragoza-CITA, Miguel Servet 177, 50013 Zaragoza, Spain; arbaoui.amira@yahoo.fr (A.A.); belanche@unizar.es (A.B.); 2Servicio de Experimentación Animal, Universidad de Zaragoza, Miguel Servet 177, 50013 Zaragoza, Spain; ggonzalo@unizar.es

**Keywords:** beef cattle, total mixed ration, performance, rumen fermentation, methane

## Abstract

Enteric methane emissions from ruminants are considered to represent ca. 3.5% of total anthropogenic greenhouse gases (GHG), and their reduction should be dealt with in order to address the climate emergency. Enteric methane production is lower in animals consuming high-concentrate diets than in those fed high-forage diets, and mixing the components of the diet (forage and concentrate) instead of offering them separately has been claimed to reduce GHG emissions. On these grounds, the objective of this study was to assess the effects of using a total mixed ration (TMR), compared to distributing the concentrate and the forage separately, on the productive performance of beef cattle fed high-concentrate diets and on rumen fermentation characteristics, including methane production. Generally speaking, the feeding system had not a significant effect on the productive performance of the animals or on rumen fermentation characteristics. Hence, the administration of TMR diets to feedlot beef cattle vs. offering the components (forage and concentrate) separately does not seem to alleviate the emissions of enteric methane.

## 1. Introduction

The anaerobic digestion of feed and the removal of hydrogen from the rumen by methanogenic bacteria result in methane production as an end product. Enteric fermentation of ruminants is considered the main source of greenhouse gas (GHG) emissions from agricultural production [1], representing ca. 3.5% of total anthropogenic GHG [2,3]. The beef cattle sector is the major contributor in Europe [4], even though the feedlot system plays a much smaller part compared to grassland-based or mixed systems in Western Europe [5]. Besides its negative impact on the environment, the methanogenesis process represents a loss of 3–10% of the gross energy intake of the animal and leads to the unproductive use of dietary energy [6]. As reducing the ruminant population being farmed is not an option, increasing the productivity of cattle to reduce methane emissions is a key area of interest [7].

Feeding strategies such as replacing maize with barley in the concentrate [8] or reducing the particle size of the forage [9] have been investigated, and in this context, alternative feeding strategies such as total mixed ration (TMR) might be of interest [10]. The advantages and disadvantages of TMR feeding of dairy cows were presented by Schingoethe [11], but no mention was made of CH₄ emissions. In fact, very little research is available on the effect of feeding systems (e.g., component or choice feeding of forage and concentrates vs. feeding of TMR) on CH_4_ production by dairy cows [10,12]. With respect to beef cattle, the published literature seems to be even more scarce [7,13], with no data available for beef cattle fed high-concentrate diets. To our knowledge, the present paper is the first one dealing with the effect of feeding a TMR vs. feeding straw and concentrate separately on methane and CO_2_ emissions by feedlot beef cattle. Allegedly, the use of TMR has significant benefits, such as increasing feed intake and digestibility, minimising choice feeding among dietary components, and allowing sufficient fibre intake to warrant rumen health (e.g., a stable ruminal pH and a lower acetate/propionate ratio) [14,15] compared to animals fed forage and concentrate separately.

On the above grounds, it was hypothesised that (1) using TMR compared to distributing the concentrate and the forage separately may affect the methane yield; (2) using TMR compared to distributing the concentrate and the forage separately may decrease the risk of acidosis; and (3) using TMR compared to distributing the concentrate and the forage separately may improve the productive performance of the animals. As a result of the previous hypothesis, the objective of the present work was to evaluate the effects of using TMR compared to distributing the concentrate and the forage separately on the productive performance of feedlot cattle and on rumen fermentation variables, including methane and CO_2_ production.

## 2. Materials and Methods

### 2.1. Animals and Diets

The experiment was carried out, over a period of 147 days, at the commercial farm Murilló Fresh Foods SL, in the Spanish Pyrenees, and at the Servicio de Experimentación Animal (SEA) of the University of Zaragoza. In the commercial farm, one hundred sixteen Montbéliarde crossbred male calves were distributed in two pens of fifty-eight animals each, according to their weight and age. Each pen received either a control diet (CS) where barley straw (ground through a 6 mm sieve and pelleted—8 mm diameter) and concentrate were offered separately to the animals and a treatment (TMR) in which both ingredients were offered as a total mixed ration. Ground and pelleted straw was provided by NAFOSA (Monzón, Huesca, Spain).

Similar average weights and ages (267 ± 2.8 kg and 190 ± 1.1 days for treatment CS, and 268 ± 2.8 kg and 189 ± 1.0 days for treatment TMR) of the animals receiving the two treatments were sought. Animal care and handling was in accordance with the Spanish Policy for Animal Protection RD 1201/05, which meets the EU Directive 86/609 on the protection of animals used for experimental and other scientific purposes. The experimental procedure was approved by the Ethics Committee of the University of Zaragoza (code PI26/21).

Nine animals from each treatment were transported to the SEA and individually accommodated in 1.7 × 3.4 m pens with slatted concrete floors, automatic water dispensers, and two separate troughs for concentrate and barley straw. Digestibility and rumen fermentation studies were carried out at this location. One week after arrival to the SEA, a 120 mm long, 11 mm internal diameter permanent cannula was inserted in the dorsal sac of the rumen of all animals. The calves were allowed to recover from surgery for two weeks. All experimental procedures involving the animals at SEA were approved by the Ethics Committee of the University of Zaragoza (code PI57/21).

All animals consumed the same commercial concentrate (Table 1), which was formulated to have 2785 kcal/kg of metabolisable energy (ME) and 14% crude protein (CP). These values are in accordance with the requirements estimated for Montbéliarde crossbred male beef calves of 190 days of age and with a daily gain target of 1.5 kg/day [16]. Concentrate and straw, and TMR, were always offered ad libitum. The calves had free access to water from the supply network throughout the experimental period.

### 2.2. Experimental Procedures

In the on-farm trial, concentrate and TMR were offered through a continuous feeding system, and straw was made freely available in the fodders. Animals were weighed on a custom-made scale (Básculas Costa, S.L., Fraga, Huesca, Spain) at the beginning of the trial and then every four weeks at 0800 h. The average daily gain (ADG) was estimated as the regression coefficient of individual live weight (LW) on time.

At the SEA, concentrations of treatment CS and TMR were offered once a day (0800) in quantities enough to allow at least 10% daily refusals. The straw was offered, in a separate trough, three times daily (0900, 1200, and 1800) to ensure ad libitum access. Animals were weighed on a custom-made scale built at the own SEA at the beginning of the trial and then every four weeks prior to feed distribution. In this location, a daily record of the amounts of TMR and of concentrate and straw from treatment CS offered to each individual animal was carried out along the whole trial. Representative samples of the concentrate, TMR, and straw were obtained weekly throughout the experimental period, pooled on a three-week basis (n = 7), and analysed for chemical composition Table 1. Concentrate and TMR refusals were also collected daily, whereas straw refusals were collected weekly. Refusals of concentrate, TMR, and straw were pooled on an animal basis. Feeds and refusals were dried at 104 °C for 24 h to determine individual dry matter intake (DMI).

On day 15 of the experimental period, samples of ruminal fluid were taken, as previously described [8], from the animals at SEA at 0 (before the administration of the concentrates), 3, 6, and 9 h after feed offer (0800, 1100, 1400, and 1700, respectively). The pH was measured immediately using a portable pH meter (model Seven2GO, Mettler-Toledo AG, Schwerzenbach, Switzerland). Aliquots of rumen liquid were taken, in duplicate, for ammonia, lactic acid, and volatile fatty acids (VFA) analysis. Moreover, in the samples from hour 0, a subsample (5 mL) was taken and immediately frozen in liquid nitrogen and then stored at −80 °C until determination of the concentration of the main microbial groups. Five days after rumen sampling (on d20), assessment of gas production in the rumen was started. To this purpose, an aluminum tube (10 mm o.d. and 8 mm i.d.) was inserted through the whole length of the cannula; this tube had a conic rubber gasket to allow the sealing of the fistula and avoid the entrance of air into the rumen. The outer end of the tube was connected to a gasometer of known section of the column, made expressly for the experiment. Only when the pressure in the column was positive (the plunger was rising, and therefore it was certain that gas was escaping from the rumen) was a sample (20 mL) of the gas in the tube obtained with a syringe and immediately injected, at a constant pressure (using a customised device), into an infrared sensor (IR15TT-R, SGX Sensortech, Katowice, Poland) previously calibrated with known concentration standards of CH₄ and CO_2_. Afterwards, the column was allowed to equilibrate, and gas production was assessed for 6 min by measuring the height of the plunger. In the case of air swallowing during the measurements, which led to a very low concentration of gases in the rumen, the process was repeated. Gas production and sampling were carried out at 0700 (before administration of the concentrate or the TMR), 1100, and 1900. As the process was time-consuming, only 3 calves were sampled at each time (same animals within the day of the process, repeating with new animals every next day). Rumen liquid sampling was repeated on d27, and gas sampling was repeated on d28.

The feed intake pattern of TMR and of concentrate and straw in animals fed CS was assessed on day 26 of the experimental period in the animals at SEA. Recording intervals were 0800–1200, 1200–1600, 1600–2000, and 2000–0800 of the next day. The procedure was repeated on day 37. On days 33–36, digestibility was estimated, also in the animals at SEA, using ashes insoluble in hydrochloric acid (AIA) as a marker. To this purpose, spot faecal samples (50–100 g) were taken directly from the rectum at 8:00 a.m. (just after feed distribution) and 17:00 p.m. during the four consecutive days. Samples of compound feed, TMR, and straw were taken the week where digestibility was assessed, and faecal samples were dried at 60 °C for 48 h and ground to pass through a 1 mm sieve for AIA analysis. Spot samples of faeces and refusals were also analysed for dry matter (DM), organic matter (OM), CP, and neutral detergent fibre (NDF) content.

All animals (from SEA and commercial farms) were slaughtered when they reached ca. 500 kg LW, and dressing percentage (cold carcass weight as a proportion of live weight) and carcass classification (according to the SEUROP classification system [17]) were obtained from the slaughterhouse.

### 2.3. Chemical Analysis

The determination of DM (ref. 934.01), OM (ref. 942.05), CP (ref. 976.05), and ether extract (EE; ref. 2003.05; only in feedstuffs) in samples of feedstuffs, feed refusals, and faeces previously ground through a 1 mm sieve was carried out following the AOAC [18] procedures. The concentration of NDF, acid detergent fibre (ADF), acid detergent lignin, and total starch in the concentrates was analysed as previously described [8]. Concentration of AIA in feedstuffs and faeces was assessed using the procedure A described in [19].

Ammonia concentration was determined colorimetrically [20], and VFA by gas chromatography [21]. Total lactic acid concentration was measured using a colorimetric method [22].

### 2.4. DNA Extraction and Microbes’ Abundance

Samples were freeze-dried, thoroughly mixed, and disrupted (Mini-Bead Beater, Biospec Products, Bartlesville, OK, USA). The microbial DNA was extracted using the Qiagen QIAmp DNA Stool Mini Kit (Qiagen Ltd., West Sussex, UK) following the manufacturer’s recommendations, except that samples were initially heated at 95 °C for 5 min to maximise bacterial cell lysis. The concentration and purity of extracted DNA were tested in Nanodrop ND-1000 (Nano-Drop Technologies, Inc., Wilmington, DE, USA). An abundance of total bacteria, methanogens, protozoa, and anaerobic fungi was assessed by qPCR at the Servicio de Secuenciación y Genómica Funcional of the University of Zaragoza, as previously described [23]. Primer sets used were as follows: 16S rRNA forward GTGSTGCAYGGYTGTCGTCA and reverse ACGTCRTCCMCACCTTCCTC for total bacteria; the mcrA gene forward TTCGGTGGATCDCARAGRGC and reverse GBARGTCGWAWCCGTAGAATCC for methanogenic archaea; the 18S rRNA forward GAGGAAGTAAAAGTCGTAACAAGGTTTC and reverse CAAATTCACAAAGGGTAGGATGAT for anaerobic fungi; and the 18S rRNA forward GCTTTCGWTGGTAGTGTATT and reverse CTTGCCCTCYAATCGTWCT for protozoa. Cycling conditions consisted of an initial denaturation at 95 °C for 5 min, followed by 35 cycles of 95 °C for 15 s, 61 °C (56 °C for methanogens and protozoa) for 30 s, and 72 °C for 55 s. The absolute amount of each microbial group, expressed as DNA copies per mg DM, was determined as described in [23] using serial dilutions of known amounts of standards consisting of the plasmid pCR 4-TOPO (Invitrogen, Carlsbad, CA, USA) containing inserted gene fragments of 16S, mcrA, or 18S from each respective microbial group.

### 2.5. Mathematical and Statistical Methods

Dry matter digestibility (DMD; g/kg) was determined as:$$DMD=1-\frac{\left[AIA\right]intake}{\left[AIA\right]excreted}$$
whereas digestibility of OM (OMD), CP (CPD), and NDF (NDFD) was calculated as:$$OMD,\;CPD\;or\;NDFD=1-\frac{\left[AIA\right]intake}{\left[AIA\right]excreted}\times \frac{nexcreted}{nintake}$$
being *[AIA]_intake_* the concentration of AIA in the DM consumed by the animal (including barley straw-BS in the CS treatment), *[AIA]_excreted_* the average concentration of AIA in the spot faecal samples (DM) collected at different times (n = 8), *n_excreted_* the proportion of OM, CP, or NDF in the faeces, and *n_intake_* the proportion of those fractions in intake (including BS in the CS treatment).

Data on final LW, ADG, and dressing percentage were analysed using the PROC MIXED of SAS (SAS Inst. Inc., Cary, NC, USA, v 9.4) with treatment as the fixed effect and animal as random. A comparison between animals at the commercial farm and at the SEA was also carried out, including the housing effect in the model. The initial body weight and age were used as covariates. Two animals at the commercial farm, one from each treatment and one from treatment TMR at the SEA, died due to complications after cannulation and had to be removed from the statistical analysis. The intake of straw and concentrate and digestibility values, along with log-transformed microbial data, were also analysed using PROC MIXED, including treatment as a fixed effect and animal as random. Regression between straw and concentrate intake for CS animals was performed using the PROC REG protocol of SAS. Organic matter intake was used as a covariate for digestibility coefficients. The pattern of intake of TMR was compared to the pattern of intake of concentrate in animals fed CS using a repeated measures analysis. The MIXED procedure of SAS was used, and treatment, day of recording, time of recording nested within day, and interactions between treatment and day or time of recording were used as fixed factors, and the animal was random. As the pattern of intake of straw was only assessed in treatment CS, the repeated measures analysis included day of recording and time of recording nested within day as fixed effects and animal as random. Rumen fermentation variables (including methane production) were analysed as repeated measures with the MIXED procedure with treatment, sampling time nested within day, sampling day, and all possible interactions as fixed effects, and animal as random. Sampling time nested within day was used as a repeated measure. This statistical analysis of rumen fermentation variables is identical to that previously described by our group [8]. The relationship between the proportion of concentrate or straw in the ration and the emission of CH₄ and CO₂ (in litres/day) was analysed using the PROG REG procedure. Abundance of total bacteria, methanogens, protozoa, and anaerobic fungi was also analysed with the MIXED procedure, with treatment, sampling day, and their interaction as fixed factors, and animal as random. Sampling day was used as a repeated measure. Frequencies of the different carcass classifications were analysed as in [8]. The variance–covariance structure was selected based on the lowest Akaike information criterion on all occasions. For all data, differences were considered significant if *p* < 0.05.

## 3. Results

### 3.1. Productive Performance of the Animals

The treatment did not affect (*p* > 0.1) the final LW, ADG, dressing percentage (Table 2), or carcass classification. The most frequent categories, according to the SEUROP classification system [17], were AO2- (13.4%) and AO2 (12.5%). In this classification, A stands for males between 12 and 24 months of age, and O stands for not very good carcass conformation (medium muscular development) [17]. Number 2 indicates minor subcutaneous fat content, with muscles always apparent, and − stands for lower values within the category [17]. On the contrary, animals housed individually at the SEA showed significantly (*p* = 0.0005) higher dressing percentages than animals at the commercial farm. The interaction between the treatment and the type of housing was not statistically significant in any case.

### 3.2. Intake and Digestibility

Total DMI, expressed either as kg/day or g/kg LW^0.75^, did not vary between treatments (*p* > 0.1), and the proportion of straw consumed by the CS animals, expressed as a percentage of the total DM intake, was 10.9 ± 1.18 (s.e.m.)%. Organic matter intake did not affect digestibility values (*p* > 0.1) but digestible OM intake (DOMI; *p* < 0.05). However, no differences between treatments were found for this latter variable or for digestibility values (*p* > 0.1; Table 3). The regression between straw and concentrate intake (kg/d of straw = −0.0321 × kg/d of concentrate + 1.2393; R^2^ = 0.0191) in animals fed CS was not significant (*p* > 0.1).

Regarding the pattern of intake (Figure 1), there were no significant differences (*p* = 0.75) between TMRT and concentrate in diet CS, with significant effects of recording day (*p* = 0.036; 363 vs. 416 g DMI/h for days 1 and 2, respectively) and recording time interval (*p* < 0.0001). With respect to this latter, animals consumed higher amounts of DM in the first four hours after feed offer (35.5% and 29.1% for TMR and concentrate in diet CS, respectively), with no differences between time intervals thereafter.

Concerning the pattern of straw intake, no differences between days of recording (*p* = 0.85) or recording time intervals (*p* > 0.05) were found.

### 3.3. Rumen Fermentation and Microbial Abundance

The interaction between treatment and time of sampling within a day was not significant (*p* > 0.1) for any of the variables of rumen fermentation, including gas production. However, the interaction between treatment and day of sampling was significant (*p* = 0.0001) for ammonia concentration, with higher values for CS than for TMR on the first day of sampling (53.0 vs. 29.4 mg/L), but the opposite on the second day (29.4 vs. 97.4 mg/L).

The effect of sampling day was significant for the molar proportions of propionate (*p* = 0.0295) and butyrate (*p* = 0.0441). Propionate proportion decreased from 44.3 mmol/100 mmol on day 1 to 40.5 mmol/100 mmol on day 2, whereas butyrate proportion increased from 12.1 to 14.5 mmol/100 mmol. Sampling day also had an effect on the percentage of CO_2_ in the gas produced in the rumen (*p* = 0.0014; 76.2 vs. 65.3% for day 1 and day 2, respectively). The sampling time within a day had an effect on pH (*p* < 0.0001), molar proportion of propionate (*p* = 0.014), and hourly production of CO_2_ (*p* = 0.0132). Values of pH were always higher before feeding than at the rest of the sampling times, whereas propionate proportion varied between sampling times only during the first sampling day (lower values before feeding, higher values at 3 and 6 h after feeding, and intermediate values at 9 h after feeding). When the hourly production of CO_2_ was analysed within a determined day, there were no differences between sampling times (*p* > 0.05). It is worth mentioning that the sum of the percentages of CH_4_ and CO_2_ was 88.5% and 87.6% for CS and TMR, respectively (*p* = 0.78), with higher values for the first than for the second day of sampling (93.7% vs. 82.2%; *p* = 0.006).

The treatment effect was significant only for rumen pH (*p* = 0.041) with higher values for CS compared to TMR animals (5.87 vs. 5.58; Table 4). None of the other variables related to rumen fermentation, including gas production (Table 5), were affected by the diet.

In animals fed CS, increasing the proportion of concentrate in the ration had no effect on CH_4_ or CO_2_ production (litres/day; *p* > 0.1).

The sampling day had an effect on the abundance of protozoa (*p* = 0.022) and the ratio of methanogens/10^6^ bacteria (*p* = 0.048), with lower values for the first than for the second day in both cases. Diet type did not affect the abundance of total bacteria, methanogens, protozoa, and anaerobic fungi, or the ratio of methanogens/10^6^ bacteria (*p* > 0.05; Table 6).

## 4. Discussion

### 4.1. Intake, Digestibility, and Animal Performance

In dairy cows, TMR feeding has been alleged to have a series of advantages [11,12,14], including no choice among feeds and more efficient digestion and utilisation of the diet because of increased uniformity of substrate for microbial activity in the rumen. Also in dairy cows, the published literature reports no differences in intake between TMR vs. separate feeding of forage and concentrate at ratios varying between 39:61 and 60:40 [12,14]. Regarding beef cattle, giving a mixed ration containing rolled barley (31.3%), rolled corn (31.3%), corn silage (15.5%), and alfalfa hay (18.9%) (35:65 forage to concentrate ratio) or a free choice among those ingredients offered individually did not affect intake or ADG [24]. No differences in DMI and nutrient digestibility between TMR and ingredients offered separately have also been reported in Holstein steers fed forage and concentrate at ratios of 27:73 [7] or in Hanwoo steers fed 15% forage and 85% concentrates [25]. This lack of differences in intake between TMR and separate feeding, regardless of the proportion of forage and concentrate in the diet, might indicate that, at those forage-to-concentrate ratios, the limitation for intake was not physical but due to the concentration of metabolites such as VFA in blood and organs like the liver, for instance. In the present experiment (10% barley straw and 90% concentrate), there were also no differences in intake or ADG between animals offered CS or TMR. Average intake (Table 3) of concentrate (8.09 kg/day) in CS animals was only 7% higher than previous values (7.58 kg/day) obtained with feedlot cattle of the same breed but slightly older (208 days) and heavier (314 kg at the beginning of the experiment) [8]. Intake of pelleted straw in animals fed CS (0.990 kg/day and 11.2 g/kg LW^0.75^, representing 10.9% of total DM intake) was roughly in accordance with the values predicted by the INRA [16] for animals of similar characteristics (9.6% of barley straw in the intake). The 14% higher values in our experiment may be explained by the fact that grinding reduces the particle size of the straw, which is expected to increase ruminal passage rate [26], hence increasing DMI, especially for low-quality forages [27]. Unfortunately, no published papers have been found dealing with the effects of reducing the particle size of straw on intake by beef cattle fed a high-concentrate ration. It is worth mentioning that the percentage of pelleted barley straw included in the TMR (10%) nearly matched that consumed by the CS animals. The percentage of straw consumed by the animals in our experiment was very similar to that previously recorded (10.2%) in slightly older (208 days) and heavier (314 kg) animals [8].

Regarding the pattern of intake (Figure 1), the lack of differences between TMR and concentrate in the CS diet can be easily explained by the fact that TMR included only 10% barley straw. The higher intake of both TMR and concentrate in the CS diet during the first 4 h after feed offer is in accordance with previous results obtained in beef calves fed a high-concentrate diet [28]. As straw in the CS diet was offered three times daily, there were no differences in its pattern of intake across recording time intervals.

Despite the allegedly more efficient digestion and utilisation of the TMR diet compared to the separate feeding of the forage and concentrate in dairy cows [12], digestibility values from the present trial (Table 3) were comparable to those previously obtained using similar animals and diets [8,29,30] and were not affected by the treatment. The lack of differences in digestibility between TMR and separate feeding has previously been reported in Hanwoo steers fed 15% oat hay and 85% concentrates [25] and supports the idea of non-different rumen conditions between treatments. Rumen conditions in the present experiment, including fermentation variables (Table 4), gas emissions (Table 5), and microbial abundance (Table 6), were not affected by the feeding system (TMR or CS). Average digestibility values found in the present experiment were ca. 4% lower than those previously reported [8] for animals of the same breed, housed in the same farm, and fed a high-concentrate diet based on maize (455 g/kg concentrate) and barley (155 g/kg concentrate). It must be taken into account that these authors performed in vivo digestibility trials, whereas in the present experiment digestibility was estimated using AIA as a marker.

As a result of the lack of differences in intake and digestibility, no effect of the treatment on animal performance (Table 2) was expected. The lack of differences in ADG has previously been reported in fattening calves consuming 34.4% forage and 65.6% concentrate [24], or in Hanwoo steers consuming 64% forage and 36% concentrate during the growing period (7–14 months of age) [31]. Average dressing percentage was numerically higher in this trial than in that reported previously by our group (54.8% vs. 53.8%) [8], but no differences were detected between diets CS and TMR in this experiment. This lack of differences in dressing percentage between TMR and ingredients offered separately has recently been reported [31]. Surprisingly, dressing percentage was higher in animals housed at the SEA than in animals at the commercial farm. It can be speculated that animals at the commercial farm had more difficulties in readapting their social relationships with their mates, leading to a higher stress [32,33]. However, this stress should have affected ADG, not dressing percentage [33].

### 4.2. Rumen Fermentation and Microbial Abundance

Concentration of treatment CS and TMR were fed once daily ad libitum, and it is recognised that offering the straw also once daily would have been a better practice than feeding the straw three times daily. However, it was observed that offering all the straw required daily just once led to an important contamination of the troughs with saliva. This contamination was not observed in the concentrate troughs, probably because mastication (and hence saliva production) was not as important as with straw (even ground and pelleted). Our decision was then to fill the straw containers three times daily, making sure that the roughage was always available in quantities enough to satisfy the animals’ demand. This way the animals had access to straw freely throughout the day, and we do not consider that the feeding behaviour would have been considerably affected if the straw had been offered once daily. Hence, the possible effects of the straw feeding frequency on ruminal pH stability, fermentation kinetics, and microbial activity were disregarded.

Even though feeding TMR vs. separate ingredients has been claimed to reduce the risk of acidosis [34] or other metabolic disorders [35] in dairy cows, key papers have been published reporting no differences in rumen pH due to feeding system [14]. The lack of differences in rumen pH due to the feeding system has also been reported in Holstein steers fed 73% concentrates and 27% roughage [7]. To our knowledge, there is no published literature dealing with the effect of feeding systems on rumen pH in beef cattle consuming 90% concentrates. In the present experiment, daily average rumen pH in animals fed TMR was close to that considered the benchmark for ruminal acidosis (5.6 or below [36,37] and lower than that in animals fed CS (Table 4). This surprising result could be explained by the fact that the TMR diet was offered once daily, whereas the straw in the CS diet was offered thrice daily. This may have had a positive effect on rumen conditions, allowing for a more stable fermentation. Lack of differences on other rumen fermentation variables (including NH_3_-N concentration, total VFA, or molar proportion of each individual VFA in the rumen liquor) due to the feeding system has also been previously reported in dairy [38] and beef [7] cattle.

The available information on the effect of feeding systems (i.e., component or choice feeding of forage and concentrates vs. feeding of TMR) on CH_4_ production by dairy cows is very limited [10,12], and results show no difference between feeding strategies. Methane production by Holstein steers fed 52–73% concentrate and 27–48% forage at 1.8% or 2.4% of LW was not influenced either by the feeding system [13,39]. The absence of an effect of the feeding system on CH_4_ production in Hanwoo steers fed 15% forage and 85% concentrate has been recently reported [25], results in accordance with those found in the present trial (Table 5). The level of feed intake in our experiment was, on average, 2.4% of LW, so the lack of differences between TMR and CS was expected. However, when Holstein steers were fed 73% concentrates and 27% roughage as TMR vs. separate feeding at 1.5% LW, animals fed the TMR option produced more methane than those fed the ration ingredients separately [7]. Increasing DMI has been reported to either increase [40] or decrease [41] methane production in ruminants. The reason put forward for an increase is the higher amount of fermentable substrate [40], whereas the faster rate of passage through the rumen has been held responsible for the decrease [42]. However, in the experiments reported above [7,13,39], there were no differences in DMI between animals fed TMR or separate feeding; hence, the reasons behind the appearance of differences in CH_4_ production between TMR and feeding separately the forage and the concentrate at low levels but not at high levels of intake should be investigated.

With respect to microbial abundance, one of the objectives of this experiment was to assess whether using TMR compared to distributing the concentrate and the forage separately may affect the CH_4_ emissions. Moreover, the rumen fermentation pattern (VFA production) and the abundance of the main microbial groups (including bacteria, protozoa, anaerobic fungi, and methanogens) were assessed in order to determine if this potential CH_4_ change was associated with a shift in the rumen feed fermentation or to a variation in the rumen concentration of those microbes involved in rumen methanogenesis (mostly methanogens and protozoa). In the present study, it was noted that the rumen concentration of total bacteria, methanogens, and protozoa (Table 6) was unaffected by the treatments, which is in agreement with a lack of differences in AGV molar proportions (Table 4). In previous studies [7,25], the feeding system (TMR vs. separate feeding) also had no effect on rumen microbes. It has been demonstrated [43] that ryegrass hay promotes a similar rumen bacterial concentration as fresh ryegrass but with a higher bacterial diversity given the higher difficulty of the former to be digested. Moreover, the aforementioned study revealed greater diversity induced by the forage preservation method in the solid than in the liquid-associated microbes. The lack of analysis of the solid-associated microbiota in the present study could justify the lack of differences in the microbial concentrations. Rumen protozoal concentration is sensitive to low pH values [44], and rumen protozoa are actively involved in the H_2_ production and in the bacterial predation; hence, their decrease in the rumen could provide a positive effect by decreasing CH_4_ emissions and increasing the efficiency of N utilisation in the rumen [45]. However, these effects, and the associated increase in production performance, were not noted in this study. The high similarity between control and treatment diets must be taken into account. The relationship between the numbers of methanogens and the amount of CH₄ produced has been a topic of debate [46], and the lack of differences between TMR and CS treatments on the numbers of methanogens and protozoa is consistent with the similar CH₄ production in animals fed either diet.

## 5. Conclusions

Contrary to our first hypothesis, using TMR compared to distributing the concentrate and the forage separately does not affect methane emissions in beef cattle fed high-concentrate rations. In the conditions of this work, the risk of acidosis was marginally increased with the TMR diet, rather than decreased, which also contradicts our second hypothesis. Animal performance was not improved with the use of TMR vs. separate feeding of straw and concentrate; hence, our third hypothesis was contradicted too. The cost of TMR processing would act against the profitability of the farms, so its use in feedlot cattle fed high-concentrate rations is not recommended.

## Figures and Tables

**Figure 1 animals-15-00723-f001:**
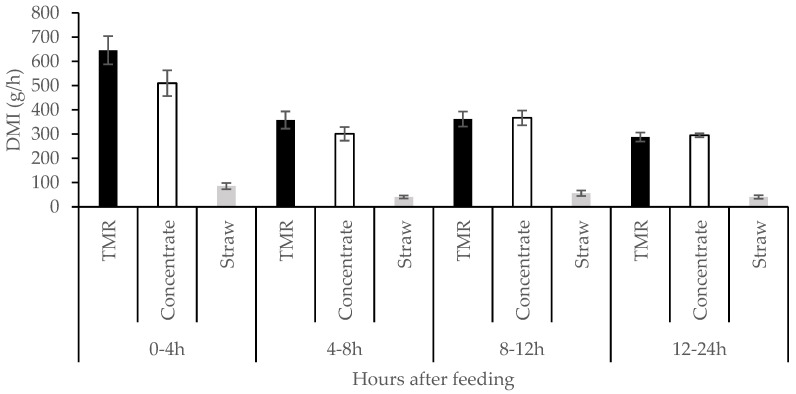
Rate of intake (dry matter; DMI) by individually housed feedlot beef cattle of a total mixed ration including 90% concentrate and 10% pelleted barley straw (TMR), and of a concentrate and pelleted barley straw, both offered separately ad libitum. Bars represent the standard error of the mean.

**Table 1 animals-15-00723-t001:** Ingredients and nutrient composition of the experimental diets.

	Concentrate	Pelleted Barley Straw	TMR ^1^
Ingredients (as fed basis). g/kg			
Maize	557.2		
Barley	40		
Maize DDGs	150		
Wheat middlings	20		
Maize gluten feed	135		
Soybean hulls	28		
Soy meal (470 g CP/kg fresh matter)	20		
Palm oil	15		
Urea	4.9		
NaCl	1.5		
Calcium carbonate	19.4		
Sodium bicarbonate	5.0		
Magnesium oxide	2.0		
Vitamin–mineral premix ^2^	2.0		
Nutrient composition (g/kg DM ± SEM; n = 7)			
OM	940 ± 1.1	938 ± 0.5	938 ± 0.6
CP	154 ± 4.2	40.2 ± 0.63	146 ± 2.9
EE	56.6 ± 3.90	9.30 ± 0.068	55.5 ± 4.71
Starch	410 ± 17.3	-	370 ± 10.2
NDF	220 ± 18.2	808 ± 5.7	260 ± 10.9
ADF	75.8 ± 9.28	460 ± 6.2	95.9 ± 7.29
Lignin	4.86 ± 0.846	43.6 ± 3.60	5.89 ± 0.480

CP: crude protein; DM: dry matter; SEM: standard error of the mean; OM: organic matter; EE: ether extract; NDF: neutral detergent fibre; ADF: acid detergent fibre; ^1^ total mixed ration, including 90% concentrate and 10% pelleted barley straw; ^2^ vitamin–mineral premix declared composition (per kg): 3.5 × 10^6^ IU vitamin A, 0.75 × 10^6^ IU vitamin D_3_, 5 g vitamin E (alpha-tocopherol acetate), 2.5 g Fe sulphate monohydrate, 250 mg K-iodate, 250 mg Co acetate tetrahydrate, 1 g Cu sulphate pentahydrate, 15 g Mn oxide, 20 g Zn oxide, 100 mg Na selenite, 150 mg butylated hydroxytoluene, and 260 g sepiolite.

**Table 2 animals-15-00723-t002:** Initial (ILW) and final (FLW) live weights, average daily gain (ADG), and dressing percentage (DP) of feedlot beef cattle consuming either a concentrate plus pelleted barley straw, both offered ad libitum (CS), or a total mixed ration including 90% concentrate and 10% pelleted barley straw (TMR).

Treatment (T)	CS		TMR		SEM	*p*-Value		
Housing (H)	B	I	B	I		T	H	T × H
ILW (kg)	266	267	268	268	8.5	0.81	0.97	0.90
FLW (kg)	484	485	486	481	9.1	0.93	0.80	0.64
ADG (kg/day)	1.54	1.62	1.58	1.56	0.063	0.89	0.52	0.27
DP (%)	53.0	57.2	53.0	55.9	1.34	0.50	0.0005	0.52

B: animals in barns at the commercial farm; I: animals in individual crates at the Veterinary Faculty of Zaragoza (Spain); SEM: standard error of the mean.

**Table 3 animals-15-00723-t003:** Dry matter intake (dry matter-DM), digestibility coefficients (g/100 g DM) of DM (DMD), organic matter (OMD), crude protein (CPD), neutral detergent fibre (NDFD), and digestible organic matter intake (DOMI) in individually housed feedlot beef cattle consuming either a concentrate plus pelleted barley straw, both offered ad libitum (CS), or a total mixed ration including 90% concentrate and 10% pelleted barley straw (TMR).

Treatment	CS	TMR	SEM	*p*-Value
Dry matter intake				
(kg/day)	9.08	8.90	0.559	0.75
(g/kg LW^0.75^)	103	102	5.2	0.79
DMD	70.4	69.9	4.37	0.92
OMD	71.5	71.2	4.26	0.93
CPD	71.4	72.0	4.37	0.90
NDFD	34.5	34.4	9.57	0.99
DOMI				
(kg/day)	6.02	6.00	0.341	0.95
(g/kg LW^0.75^)	68.5	68.3	3.93	0.96

SEM: standard error of the mean; LW: live weight.

**Table 4 animals-15-00723-t004:** Average daily rumen pH, daily average concentrations of volatile fatty acids (VFA; mg/L), ammonia (mg/L), lactic acid (mg/L), and daily average molar percentage (mmol/100 mmol) of the main VFA in the rumen fluid of feedlot beef cattle consuming either a concentrate plus pelleted barley straw, both offered ad libitum (CS), or a total mixed ration including 90% concentrate and 10% pelleted barley straw (TMR).

Treatment	CS	TMR	SEM	*p*-Value
pH	5.87	5.58	0.130	0.0410
Total VFA	128	136	7.7	0.3511
Ammonia	64.4	63.4	1.03	0.9408
Lactic acid	74.0	79.4	7.75	0.5021
Acetate	40.9	37.6	2.53	0.2089
Propionate	43.0	41.8	4.21	0.7819
Butyrate	11.6	15.0	2.76	0.2358
Iso-butyrate	0.733	0.605	0.134	0.3533
Valerate	1.38	1.01	0.382	0.3451
Iso-valerate	2.37	3.97	1.376	0.2635

SEM: standard error of the mean.

**Table 5 animals-15-00723-t005:** Percentage of methane (CH_4_) and CO_2_ in the gas produced in the rumen of feedlot beef cattle consuming either a concentrate plus pelleted barley straw, both offered ad libitum (CS), or a total mixed ration including 90% concentrate and 10% pelleted barley straw (TMR), average hourly production (L/h), weighed daily production (L/d), production per kg of digestible organic matter intake (DOMI; L7kg DOMI), and production per kg weight gain (L/kg Δ weight).

Treatment	CS	TMR	SEM	*p*-Value
% CH_4_	17.6	16.9	1.47	0.6603
% CO_2_	70.9	70.5	3.04	0.9065
L CH_4_/h	8.87	6.77	1.951	0.2910
L CO_2_/h	35.9	27.7	7.49	0.2814
L CH_4_/d	211	161	48.4	0.3135
L CO_2_/d	857	657	186.4	0.2955
L CH_4_/kg DOMI	37.1	28.0	8.73	0.3102
L CO_2_/kg DOMI	150	115	34.0	0.3114
L CH_4_/kg Δ weight	127	154	46.4	0.5616
L CO_2_/kg Δ weight	511	629	177.0	0.5111

SEM: standard error of the mean.

**Table 6 animals-15-00723-t006:** Abundance (log of number of rDNA copies/mg DM) of total bacteria (Bact), methanogens (Met), protozoa (Prot), and anaerobic fungi (Fung) in the rumen of feedlot beef cattle consuming either a concentrate plus pelleted barley straw, both offered ad libitum (CS), or a total mixed ration including 90% concentrate and 10% pelleted barley straw (TMR), and the ratio Met/10^6^ Bact (ΔC_T_).

Treatment	CS	TMR	SEM	*p*-Value
Bact	9.75	9.82	0.078	0.3727
Met	6.45	6.16	0.226	0.2163
Prot	5.73	5.37	0.814	0.6628
Fung	3.90	3.52	0.525	0.4823
ΔC_T_	197	125	48.4	0.1594

SEM: standard error of the mean.

## Data Availability

The original contributions presented in the study are included in the article, further inquiries can be directed to the corresponding author.

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
