# Peer review of "Unmixed Diet Versus Total Mixed Ration in Beef Cattle Fed High-Concentrate Diets: Effects on Methane Emissions, Animal Performance, and Rumen Fermentation"

_animals, 2025, doi:10.3390/ani15050723_

Round 1

Reviewer 1 Report

Comments and Suggestions for Authors

The fundamental problem with the research was the field experiment in the commercial farm. Two pens, one for TMR and other for CS. Animals within a pen are observational units and not experimental units. Therefore, they are false replications.
The animals that were separated for the metabolism assays are proper experimental units. Therefore, I suggest the removal of the field part of the experiment, and focus strictly on the metabolism assay.
In this case, I suggest a major revision.

Author Response

Dear reviewer,

Thank you very much for your encouraging and supporting comments.

Reviewer: The fundamental problem with the research was the field experiment in the commercial farm. Two pens, one for TMR and other for CS. Animals within a pen are observational units and not experimental units. Therefore, they are false replications.

The animals that were separated for the metabolism assays are proper experimental units. Therefore, I suggest the removal of the field part of the experiment, and focus strictly on the metabolism assay.

In this case, I suggest a major revision.

Response: The animals at the commercial farm were only used to assess ADG, and dressing percentage and carcass classification. There were individual data for each animal, so the animal was the experimental unit (as for the metabolism assays). Comparing animal performance between calves at the commercial farm (in barns) and those at the experimental farm (individually housed) allowed us to check if the housing method had an effect in animal performance. We have tried and made this clearer (L31-33, L137-138, L166 and L174). Any way, if you still consider that the results of the commercial farm should be deleted, we are willing to do it.

A major revision has been undertaken following your suggestions and those from the other two reviewers.

Kind regards,

Antonio de Vega

Reviewer 2 Report

Comments and Suggestions for Authors

This is a surprising study, and the experimental results did not match the expected results. The experimental results are also inconsistent with traditional understanding, but these do not affect its research value. This requires the author to provide more experimental details to ensure the accuracy and credibility of the results of this study.
1. The author should further supplement similar research progress in the preface to see if there are similar research results. If not, the novelty of this study can be highlighted.
2. Please further supplement the measurement methods for each indicator, especially for indicators such as feed intake and gas emissions. In the Materials and Methods section, the author should provide every detail to ensure that readers have a clear understanding of the experimental process.
3. In the discussion section, the author should explain why there is a discrepancy with traditional understanding, from multiple perspectives such as digestive physiology and gut microbiota.
Although the results of this study are inconsistent with traditional understanding, they still have important value, and whether they can be published depends on the author's clear explanation of each experimental detail and the reasons behind the results.

Author Response

Dear reviewer,

Thank you very much for your kind and useful comments; please find my answers below.

Reviewer: This is a surprising study, and the experimental results did not match the expected results. The experimental results are also inconsistent with traditional understanding, but these do not affect its research value. This requires the author to provide more experimental details to ensure the accuracy and credibility of the results of this study.

  1. The author should further supplement similar research progress in the preface to see if there are similar research results. If not, the novelty of this study can be highlighted.
  2. Please further supplement the measurement methods for each indicator, especially for indicators such as feed intake and gas emissions. In the Materials and Methods section, the author should provide every detail to ensure that readers have a clear understanding of the experimental process.
  3. In the discussion section, the author should explain why there is a discrepancy with traditional understanding, from multiple perspectives such as digestive physiology and gut microbiota.
    Although the results of this study are inconsistent with traditional understanding, they still have important value, and whether they can be published depends on the author's clear explanation of each experimental detail and the reasons behind the results.

Response: Published literature dealing with the effect of feeding TMR vs. feeding forage and concentrate separately on methane production by beef steers is really scarce (L60-62), with no information obtained in feedlot cattle fed 10% straw and 90% concentrate. Previous results seemed to be dependent on feeding level (as discussed in L464-475).

  1. The originality of our paper has been highlighted in L62-64.
  2. The way feed intake was recorded is described in L124-136, and gas measurements have been detailed in L144-161. We can upload a video showing the process of gas measurement and sampling as supplementary material, if you consider it worthy.
  3. As stated above, very few papers have been published dealing with the effect of feeding TMR vs. feeding forage and concentrate separately on methane production by beef steers. We have tried to take into account all variables you mention when writing the Discussion section which has been considerably modified.

Kind regards,

Antonio de Vega

Reviewer 3 Report

Comments and Suggestions for Authors

The study design requires refinement, particularly regarding feeding frequency—TMR was fed once daily, while straw was fed three times daily, potentially influencing ruminal pH and fermentation independently. The TMR composition (90% concentrate, 10% straw) is unbalanced, as typical TMR diets have higher forage inclusion. High standard errors (SEM) suggest significant data variability, indicating the need for improved precision in methodology. Additionally, the discussion section is underdeveloped and requires further elaboration. The similarity of 40% is concerning and should be addressed to ensure originality.

Author Response

Dear reviewer,

Thank you very much for your encouraging and useful comments; please find my answers below.

Reviewer: The study design requires refinement, particularly regarding feeding frequency—TMR was fed once daily, while straw was fed three times daily, potentially influencing ruminal pH and fermentation independently. The TMR composition (90% concentrate, 10% straw) is unbalanced, as typical TMR diets have higher forage inclusion. Additionally, the discussion section is underdeveloped and requires further elaboration. The similarity of 40% is concerning and should be addressed to ensure originality.

Response: TMR was fed once daily but ad libitum (L124-125). Concentrate of the CS treatment was also offered ad libitum once daily (L124-125). During the first days of the experimental period, it was observed that offering all the straw required daily just once lead to an important contamination of the troughs with saliva, so our decision was to fill the containers three times daily. This way, we assured ad libitum feeding of the straw but also a reduction in the contamination with saliva. Regarding the TMR composition, those with higher forage proportion are typical for dairy cows but not for beef cattle. In previous papers (https://doi.org/10.1016/j.anifeedsci.2014.11.008, https://doi.org/10.3390/ani13193016) published by our group,  the percentage of straw in feedlot cattle varied between 7% and 10% of the total dry matter intake, varying with animals’ age and breed. These variations are also considered by the INRA (please see L382-385). On these grounds, we decided to make a TMR containing 10% barley straw and 90% concentrate. The results obtained when feeding straw and concentrate separately (10.9 % straw, L271) indicate that our decision was correct. The SEM were usually below 10% of the average, which is not bad for beef cattle. Higher variations were found for NDF digestibility (28%, Table 3), and the proportions of butyrate (21%), iso-butyrate (20%), valerate (32%) and iso-valerate (43%) in the rumen liquid (Table 4). It must be taken into account, however, that those four volatile fatty acids were present in very low concentrations, and that a small analytical error may lead to high differences in concentration when expressed proportionally (as an example, the difference in iso-valeric concentration between CS and TMR was only 1.6 mg/L, but 68% higher with TMR. The errors for gas emissions (Table 5) were roughly around 30% of the average, and this may indicate that the methodology was inappropriate but also important individual variations. In fact, the genetic selection of cattle with low gas emissions is based on this individual variability. The discussion has been more developed, and the syntax changed to avoid similarity.

Kind regards,

Antonio de Vega

Round 2

Reviewer 2 Report

Comments and Suggestions for Authors

The author has fully responded to the review comments and made revisions, and can consider publishing.

Comments on the Quality of English Language

It is fine, and published is OK.

Author Response

Dear reviewer,

Thank you very much for your kind comments; please find my answers below.

Reviewer: Comments and Suggestions for Authors: The author has fully responded to the review comments and made revisions, and can consider publishing.

Comments on the Quality of English Language: It is fine, and published is OK.

Response: Thank you very much for your decision, which makes us really happy.

Kind regards,

Antonio de Vega

Reviewer 3 Report

Comments and Suggestions for Authors

I appreciate the authors' efforts in revising the manuscript; however, some aspects require further clarification. The difference in feeding frequency could have influenced rumen pH stability, fermentation kinetics, and microbial activity, and providing additional details on how this factor was managed. The 90:10 concentrate-to-straw ratio is lower in forage than standard recommendations, and as the cited references refer to ad libitum forage feeding. Additionally, the similarity index (33%) remains high, and refining certain sections to better highlight the novel aspects of this study compared to previous work would improve clarity. Addressing these points would enhance the manuscript’s scientific impact.

Author Response

Dear reviewer,

Thank you very much again for your encouraging and useful comments; please find my answers below.

Reviewer: I appreciate the authors' efforts in revising the manuscript; however, some aspects require further clarification. The difference in feeding frequency could have influenced rumen pH stability, fermentation kinetics, and microbial activity, and providing additional details on how this factor was managed. The 90:10 concentrate-to-straw ratio is lower in forage than standard recommendations, and as the cited references refer to ad libitum forage feeding. Additionally, the similarity index (33%) remains high, and refining certain sections to better highlight the novel aspects of this study compared to previous work would improve clarity. Addressing these points would enhance the manuscript’s scientific impact.

Response: TMR, and concentrate of treatment CS, were fed once daily ad libitum (L124-125). We recognise that offering the straw also once daily ad libitum would have been a better practice than feeding the straw three times daily. However, and as explained in the previous review-round, it was observed that offering all the straw required daily just once led to an important contamination of the troughs with saliva. This contamination was not observed in the concentrate troughs, probably because mastication (and hence saliva production) was not as important as with straw (even ground and pelleted). Our decision was then to fill the straw containers three times daily, making sure that the roughage was always available in quantities enough to satisfy the animals’ demand. This way the animals had access to straw freely throughout the day, and we do not consider that the feeding behaviour would have been different if the straw had been offered once daily. This rationale has been included in the discussion section (L444-455). Regarding the concentrate-to-straw ratio, 90:10 is usual in feedlot beef cattle fed concentrate and straw ad libitum (INRA 2018 (please see L384-387), https://doi.org/10.1016/j.anifeedsci.2014.11.008, https://doi.org/10.3390/ani13193016). The similarity index has been reduced following the suggestions from the journal editor (Emily Ma). Finally, the novel aspects of our study have been highlighted (L60-64).

Kind regards,

Antonio de Vega

Round 3

Reviewer 3 Report

Comments and Suggestions for Authors

Thank you for your detailed response and for incorporating the suggested revisions.

Feeding frequency and rumen fermentation stability
The rationale for offering straw three times daily to mitigate saliva contamination is appreciated. Additionally, the continuous availability of straw is a crucial factor that supports the acceptance of this feeding strategy.

Similarity 
The similarity index has been reduced from 33% to 26%, and 10% of this similarity is derived from a single source. The author must rectify this issue.

Author Response

Dear reviewer,

Thank you very much again for your encouraging and useful comments; please find my answers below.

Reviewer: Thank you for your detailed response and for incorporating the suggested revisions.

Feeding frequency and rumen fermentation stability
The rationale for offering straw three times daily to mitigate saliva contamination is appreciated. Additionally, the continuous availability of straw is a crucial factor that supports the acceptance of this feeding strategy.

Similarity 
The similarity index has been reduced from 33% to 26%, and 10% of this similarity is derived from a single source. The author must rectify this issue.

Response: We have tried to reduce the similarity index as much as possible, but in some instances (composition of the mineral-vitamin premix, headings of tables, affiliation, common statistical procedures, etc.) it has not been possible without a loss of important information.

Kind regards,

Antonio de Vega